# Exciton polariton condensation from bound states in the continuum at room temperature

Xianxin Wu[1,2,10], Shuai Zhang[1,10], Jiepeng Song[3,10], Xinyi Deng[3], Wenna Du[1,2], Xin Zeng[1], Yuyang Zhang[1], Zhiyong Zhang[1,4], Yuzhong Chen[5], Yubin Wang[6], Chuanxiu Jiang[1,2], Yangguang Zhong[1], Bo Wu[7], Zhuoya Zhu[1,2], Yin Liang[3], Qing Zhang[3]✉, Qihua Xiong[5,6,8,9]✉ & Xinfeng Liu[1,2]✉

Exciton–polaritons (polaritons) resulting from the strong exciton–photon interaction stimulates the development of novel low-threshold coherent light sources to circumvent the ever-increasing energy demands of optical communications[1–3]. Polaritons from bound states in the continuum (BICs) are promising for Bose–Einstein condensation owing to their theoretically infinite quality factors, which provide prolonged lifetimes and benefit the polariton accumulations[4–7]. However, BIC polariton condensation remains limited to cryogenic temperatures ascribed to the small exciton binding energies of conventional material platforms. Herein, we demonstrated room-temperature BIC polariton condensation in perovskite photonic crystal lattices. BIC polariton condensation was demonstrated at the vicinity of the saddle point of polariton dispersion that generates directional vortex beam emission with long-range coherence. We also explore the peculiar switching effect among the miniaturized BIC polariton modes through effective polariton–polariton scattering. Our work paves the way for the practical implementation of BIC polariton condensates for integrated photonic and topological circuits.

Exciton–polaritons (polaritons for short), as hybrid bosonic quasi-particles resulting from the strong coupling between semiconductor excitons and microcavity photons, can undergo Bose−Einstein condensation at elevated temperatures, promising for low-threshold coherent emitters[8,9], all-optical logic circuits[10,11], and quantum simulators[12,13]. Microcavities with higher quality factors (*Q* factors) and longer coherence times are always desirous for enhanced exciton–photon coupling. Possessing theoretically infinite-high cavity

*Q* factors and peculiar non-radiative characteristics, the bound states in the continuum (BICs) exhibit great promise in vortex beam generation and topological modulation in the linear regime[4,5,14,15], and also low-threshold nanolaser[16–18], as well as polariton accumulation[19–23]. However, the potential of polaritons from BICs (BIC polaritons) for providing unrivaled nonlinearities and topological characteristics in strong interaction scenarios is still in its infancy. Recently, BIC polariton condensation has been realized in a patterned GaAs quantum well

[1]CAS Key Laboratory of Standardization and Measurement for Nanotechnology, National Center for Nanoscience and Technology, Beijing 100190, P. R. China. [2]University of Chinese Academy of Sciences, Beijing 100049, P. R. China. [3]School of Materials Science and Engineering, Peking University, Beijing 100871, P. R. China. [4]School of Physical Science and Technology, Inner Mongolia University, Hohhot 010021, P. R. China. [5]Beijing Academy of Quantum Information Sciences, Beijing 100193, P. R. China. [6]State Key Laboratory of Low-Dimensional Quantum Physics and Department of Physics, Tsinghua University, Beijing 100084, P. R. China. [7]Guangdong Provincial Key Laboratory of Optical Information Materials and Technology, Institute of Electronic Paper Displays, South China Academy of Advanced Optoelectronics, South China Normal University, Guangzhou 510006, P. R. China. [8]Beijing Innovation Center for Future Chips, Tsinghua University, Beijing 100084, P. R. China. [9]Frontier Science Center for Quantum Information, Beijing 100084, P. R. China. [10]These authors contributed equally: Xianxin Wu, Shuai Zhang, Jiepeng Song. ✉e-mail: q_zhang@pku.edu.cn; qihua_xiong@tsinghua.edu.cn; liuxf@nanoctr.cn

waveguide at cryogenic temperature (~4 K)[6,7,24]. The room-temperature operation of this topological macroscopic quantum state is fascinating, while it is a fundamental restriction for conventional III–V semiconductors with low exciton binding energies.

Lead halide perovskites with large exciton binding energies embedded in microcavities are deemed as superb polaritonic platforms for room-temperature polariton condensation[25–28]. Various proof-of-principle condensate devices, ranging from Hamiltonian simulators and quantum fluids to investigations of rich polariton physics in non-Abelian gauge fields, have been extensively explored[13,29–33]. To date, the demonstration of perovskite polariton condensation without vertical Fabry–Pérot bulky microcavities, which is crucial for practical integrated photonics seeking more designable and compact structures, remains limited[34]. On the other hand, owing to the high optical gain, easily tunable bandgap, high defect tolerance, and good processability[35–38], perovskite single crystals are of particular interest as laser gain media, particularly the recently demonstrated BIC photonic lasing[39,40]. Nevertheless, the intriguing Bose–Einstein condensation resulting from the strong coupling between excitons and BIC cavity photons has not been exploited yet in such perovskite single crystal-based structure, from which low-threshold, large nonlinearities, and topological characteristics of polariton condensates are expected to be generated at room temperature.

In this study, we demonstrate room-temperature BIC polariton condensation in perovskite artificial photonic crystal (PhC) lattice, by combining the high-quality BIC modes with stable excitons. Single-crystalline perovskite microplatelets with robust excitons enable strong coupling between BIC mode and excitons with a Rabi splitting larger than 150 meV. Long-range coherent vortex emission in the vertical direction with a low divergence angle is obtained from the BIC polariton condensates. Additionally, we successfully switch between multiple orders of BIC polariton modes with preserved topology in a miniaturized structure size through the nonlinear interactions of polariton condensates. Our work provides a platform to couple coherent polariton condensates with orbital angular momentum at room temperature, serving as a significant basis for integrated photonic and topological circuits.

## Results and discussion

Figure 1a schematically depicts room-temperature BIC polariton condensation based on a perovskite air-hole PhC lattice. All-inorganic cesium lead bromide ($CsPbBr_3$) is selected as a suitable room-temperature polaritonic material owing to its large exciton binding energy and superior optical properties[35,41]. Relatively large-area (>20 μm), single-crystalline $CsPbBr_3$ microplatelets were synthesized on silicon substrates through chemical vapor deposition, where the lattice match between $CsPbBr_3$ and silicon led to a vertically-grown feature, restricting the deposition only on the microplatelets' exposed edges (see Methods section). Thereby, this confined growth, together with a slow nucleation rate, results in $CsPbBr_3$ microplatelets with exceptional crystal quality and atomic-level smooth surfaces without grain boundaries (Supplementary Fig. S1). Moreover, the photoluminescence (PL) emission exhibits a narrow full width at half maximum (FWHM) of ~60.7 meV, and a single-exponential decay featuring a lifetime of ~6.0 ns, suggesting a low trap density and reduced non-radiative loss. The strong excitonic peak observed in the reflection spectrum convinces the robust excitons against thermal ionization at room temperature (Supplementary Fig. S2). Subsequently, the periodic air-hole PhC lattice was carved on the $CsPbBr_3$ microplatelet by the focused ion-beam (FIB) milling, with radius $r = 57$ nm, thickness

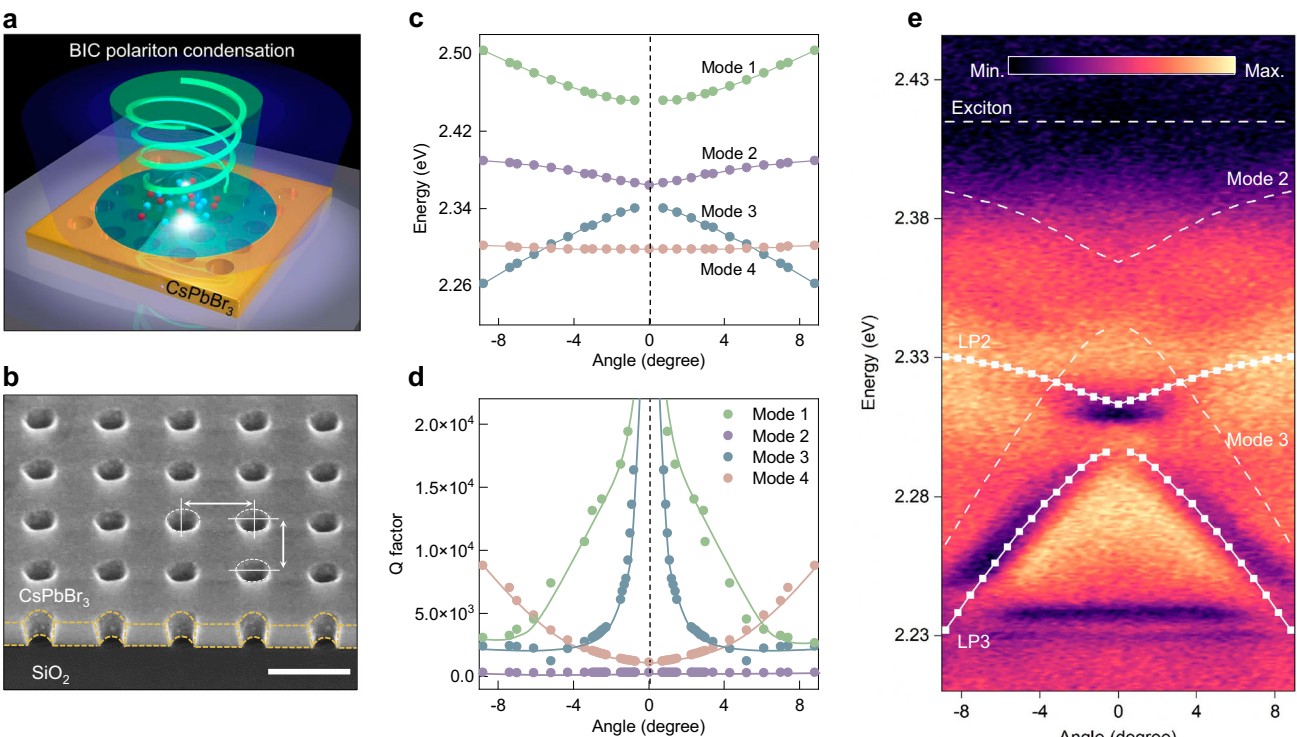

**Fig. 1 | BIC polaritons in a perovskite PhC. a** Schematic of room-temperature BIC polariton condensation from an air-hole $CsPbBr_3$ PhC lattice. A localized vortex beam is generated along the surface normal direction pumped by a non-resonant pulsed laser. **b** Tilt-view SEM image of the air-hole $CsPbBr_3$ PhC lattice. The thickness, period, and diameter of the air holes are 150, 300, and 150 nm, respectively. Scale bar: 300 nm. **c** Calculated energy-angle dispersions of the $CsPbBr_3$ PhC lattice mode in TM polarization. **d** Calculated $Q$ factors of the four PhC lattice modes plotted in **c** (Solid curves: eye-guided). The BICs are demonstrated by the near-zero linewidth and ultrahigh $Q$ factor at the normal incidence. **e** Angle-resolved reflectance spectrum of the air-hole $CsPbBr_3$ PhC lattice in TM polarization. LPB with an anti-crossing characteristic is distinguished resulting from the strong coupling between the exciton and PhC lattice mode. The fittings are based on the coupled harmonic oscillator model, and a Rabi splitting value of 158 meV is obtained.

$h = 145 \pm 5\,nm$, and periodicity $a = 290–300\,nm$ ($a = 300\,nm$ for Fig. 1b). A 30 nm polymethyl methacrylate layer was spin-coated onto the $CsPbBr_3$ microplatelet for both protection and refractive index matching. The sharp sidewalls and smooth surfaces (Supplementary Fig. S3) indicate negligible damage caused by FIB milling (see discussion of different etching conditions in Supplementary Note 1). Meanwhile, both red-shifted emission and shortened lifetime were observed in the PhC lattice compared to the pristine microplatelet ascribing to the enhanced exciton–photon coupling strength (Supplementary Fig. S2).

Figure 1c shows the calculated transverse magnetic (TM) polarized energy-angle dispersion of Bloch resonances in the PhC lattice within the gain bandwidth of $CsPbBr_3$ (see Methods section, Supplementary Note 2). Here, 0° corresponds to the high-symmetry point ($\Gamma$) of the first Brillouin zone in the square PhC lattice. Four mode dispersions stretch around the $\Gamma$-point, with two breakpoints observed in Modes 1 and 3. The faded visibility and narrowed linewidth of these two modes around the $\Gamma$-point suggest dark states with ultrahigh $Q$ factors approaching infinite, corresponding to the symmetry-protected BICs decoupled from the radiation continuum (Fig. 1d). The bright Mode 2 and top of Mode 3 possesses an energy range from 2.34 to 2.39 eV around the detection angle, in near resonance with the excitonic emission of $CsPbBr_3$ (2.41 eV). Moreover, angle and energy-resolved reflectance spectra (see Methods section, Supplementary Fig. S4) reveal similar characters of two dominated modes with decreasing curvature approaching the excitonic resonance, which matches well with the theoretically calculated dispersion of lower polariton (LP) branch, labeled as LP2 and LP3. Notably, the upper polariton (UP) branches could not be identified due to the strong absorption occurring above the exciton energy [42]. In addition, the air-hole PhC lattice, fabricated with 40 periods, exhibits a narrow mode linewidth closely resembling that of an infinite structure (Supplementary Fig. S5). A Rabi splitting energy of 158 meV and detuning $\Delta = -74.6\,meV$ could be extracted from LP3, confirming the realization of strong exciton–photon coupling (Fig. 1e). Remarkably, a similar trend of linewidth narrowing of LP3 could be observed from 18.8 meV to a minimum of 2.5 meV ($Q$ factor ~913) near the $\Gamma$-point (Supplementary Fig. S6) in measured reflectance spectrum, demonstrating the formation of symmetry-protected BIC polariton mode.

The presence of BIC polariton mode within a hybrid system provides a state with a theoretically infinite $Q$ factor, enabling ultrastrong net gain while minimizing losses in the mode dispersion, which favors polariton accumulation towards condensation. Here, femtosecond-pulsed laser excitation with an expanded square spot size of ~10 μm was applied to explore the nonlinear regime of polariton interactions (see Methods section, Supplementary Fig. S7). Figure 2a shows the angle-resolved PL spectrum of a sample with a detuning ($\Delta$) of −74.6 meV at room temperature with a low pump density of 0.6 times the condensation threshold ($P_{th}$). The dispersion is similar to that in Fig. 1d and emission is dominated at bright polariton mode LP2. As the pump density increased to $P_{th}$, a two-lobe emission occurred at the dispersion saddle point of LP3 in reciprocal space (Fig. 2b), and then dominated the entire spectrum as the pump density further increased to 1.5 $P_{th}$ (Fig. 2c), suggesting the formation of the BIC polariton condensates. This nonlinear behavior is further characterized in detail by the linear-to-superliner transition of the integrated intensity (Fig. 2d), decrease in the linewidth (Fig. 2e, Supplementary Fig. S8), and energy blueshift of the BIC polariton emission (Fig. 2f) as a function of the pump density (extracted from pump density-dependent emission spectra in Supplementary Fig. S9). Remarkably, the blueshift exhibited a larger slope below $P_{th}$ and a smaller slope above $P_{th}$, corresponding to the distinct interaction strengths of polariton–reservoir and polariton–polariton interactions, respectively [43]. As a reference, the sample with a weak negative detuning ($\Delta = -64.1\,meV$) shows a lower $P_{th}$ and larger slope of energy blueshift. This enhanced nonlinear performance

is consistent with the increase of the excitonic fraction [19], further confirming the realization of BIC polariton condensation (Supplementary Fig. S10). More results of BIC polariton condensations at different detuning are available in Supplementary Note 3. Additionally, despite the ubiquitous optical birefringence in orthorhombic $CsPbBr_3$ single crystals, BIC polariton condensation can be only observed in TM polarization, which could be attributed to the longer lifetime of the TM-polarized state resulting from a higher excitonic fraction (Supplementary Fig. S11).

Furthermore, Fig. 2g presents the time- and angle-resolved emission snapshots of BIC polariton condensation at 1.6 $P_{th}$ and three specific time points of 0.4, 0.8, and 8 ps (see Methods section and more data in Supplementary Note 4). The rapid establishment and maximum intensity of the two lobes of the BIC polariton condensate occur within 0.8 ps. Subsequently, the emission intensity gradually decreases with a continuous redshift, indicating a decrease in condensed polariton density and weakened polariton nonlinearity. Remarkably, the profile of the two lobes remains unchanged during the time evolution, and the emission always comes from the BIC polariton mode. The BIC polariton lifetime is predominantly limited by the finite cavity quality factor, which could be further alleviated through optimized fabrication processes (see discussion in Supplementary Note 4). Additionally, the BIC polariton condensation can also be realized under quasi-continuous-wave laser excitation (pulse duration: 100 ns, repetition rate: 100 kHz) at 80 K (Supplementary Fig. S12). Compared to the photonic lasing from pristine $CsPbBr_3$ microplatelets with similar sizes, BIC polariton condensation exhibits a threshold reduction of over 50%, reflecting its low-threshold characteristic and the potential for further achieving continuous optically pumped operation at elevated temperatures.

The long-range spatial and temporal coherence is another conclusive evidence for the formation of the polariton condensates, while the correlation length is constrained by the thermal de Broglie wavelength in the thermal phase [26,44]. The coherence feature was characterized using the first-order correlation function $g^{(1)}$ measured by the Michelson interferometer (see Methods section, Supplementary Fig. S13). Figure 3a illustrates the interference pattern of BIC polariton condensate emission at zero-time delay and the extracted intensity line profile along $y = 0$. Compared with the limited correlation length below $P_{th}$, clear interference fringes, extending over the entire PhC lattice, were observed above $P_{th}$. Gaussian fitting of the intensity profile yielded an FWHM of 8.7 μm, suggesting the realization of long-range spatial coherence (Supplementary Fig. S13). Moreover, temporal coherence was probed by scanning time-delayed interference patterns, where the maximum fringe visibility was observed at zero-time delay. As the delay time extended from 0 to 1.5 ps (P1–P3), the fringes gradually blurred (Fig. 3b). The temporal coherence was quantified by fitting the visibility decay from the magnitudes of the time-delayed interference patterns (Fig. 3c). The coherence time of BIC polariton condensate emission, determined by fitting a Gaussian function, was approximately one order of magnitude longer ($2.14 \pm 0.16\,ps$) than that of the referenced pump laser ($0.31 \pm 0.01\,ps$). Additionally, Fig. 3d reveals a decreasing trend of temporal coherence with increasing pump density, attributed to phase decoherence within the polariton lasing mode resulting from enhanced polariton–polariton scattering [43,45]. This trend aligns with the broadening of emission linewidth (Fig. 2e) and accelerated decay rate at higher pump density (Supplementary Note 4) above $P_{th}$. Note that longer coherence time can be achieved by further optimization of the material quality and fabrication processes (Supplementary Fig. S14). Thus, the observed long-range spatial and temporal coherences confirm the onset of room-temperature BIC polariton condensation. It should be noted that recently, polariton condensation has also been demonstrated in organic materials combining with BICs in silicon metasurfaces [46], further underscoring the broad applicability of BIC polariton condensates at room temperature.

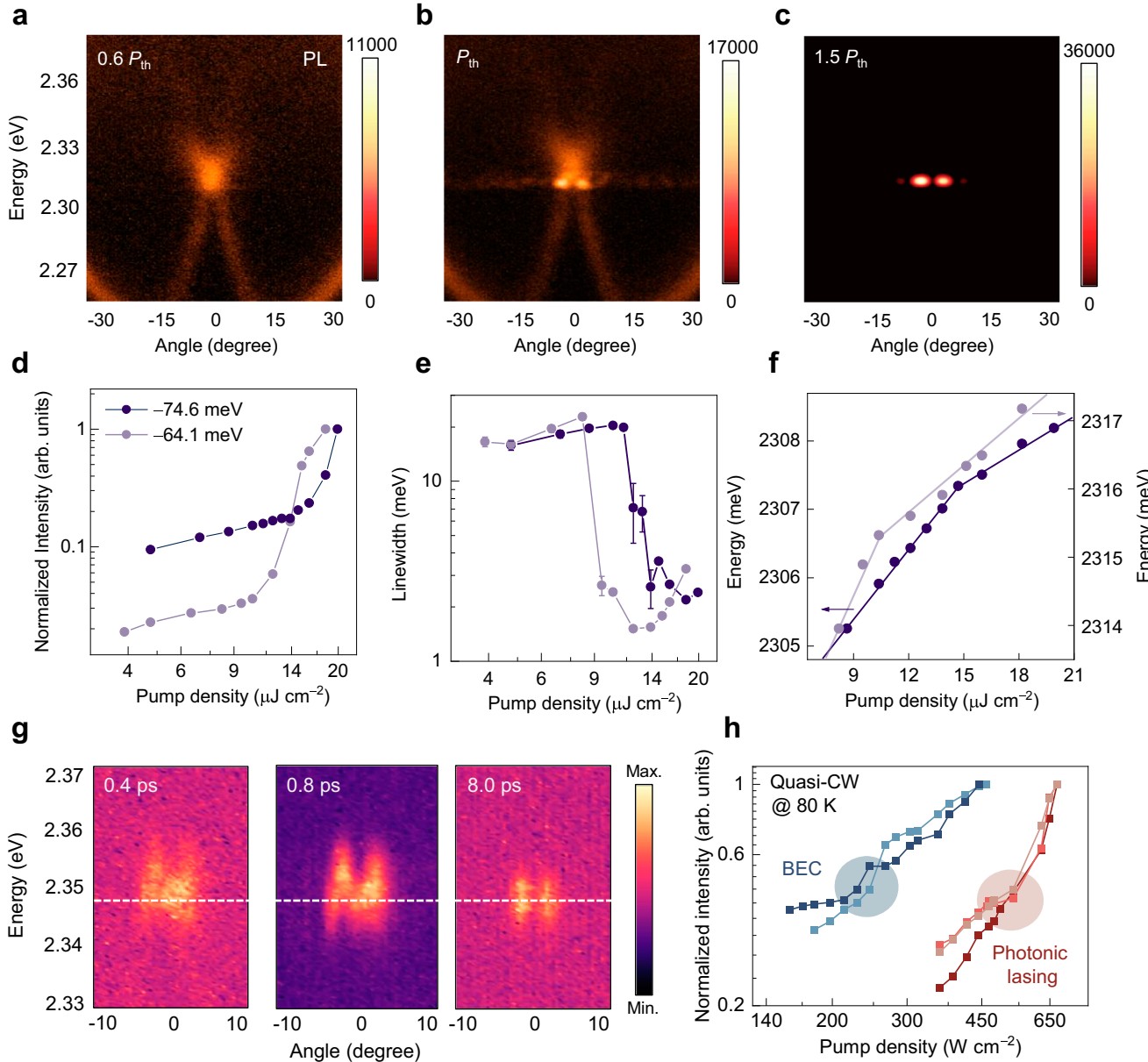

**Fig. 2 | BIC polariton condensation at room temperature. a–c** Angle-resolved PL spectra under pump densities at 0.6 $P_{th}$ (**a**): the emission exhibits a broad angular distribution (±15°); $P_{th}$ (**b**): the emission near the dispersion saddle point shows a sharp increase within a narrow angular distribution (±5°), suggesting the onset of polariton condensation; 1.5 $P_{th}$ (**c**): the state at the dispersion saddle point is massively occupied. Two samples with distinct detuning energies of −74.6 and −64.1 meV are explored. **d** Integrated emission intensity as a function of the pump density, showing a linear-to-superliner transition across $P_{th}$. **e** Linewidth as a function of the pump density, showing a clear narrowing effect across $P_{th}$. The error bar represents the standard deviation. **f** Energy blueshift as a function of the pump density, where two regimes are distinguished below and above $P_{th}$ attributing to the polariton−reservoir and polariton−polariton interactions, respectively. **g** Time-resolved and angle-resolved PL emission snapshots of an identical sample at three distinct times of 0.4, 0.8, and 8 ps, respectively. **h** Integrated emission intensity as a function of the pump density of CsPbBr$_3$ PhC lattices (left two blue lines) and pristine microplatelets (right three orange lines) with similar sizes under quasi-CW excitation at 80 K. BIC polariton condensation exhibits a threshold reduction of over 50% compared to pure photonic lasing.

Next, we investigate the inherent topological properties of the BIC polariton condensates, as the emission of vortex beams with a topological charge is anticipated, stemming from the orbital angular momentum inherited from the BIC mode[4,5]. Figure 4a, b show the back-focal plane (BFP) images of the CsPbBr$_3$ PhC lattice emission below and above $P_{th}$, respectively. Below $P_{th}$, the emission covers the entire plane of momentum space, while surpassing $P_{th}$ leads to the emergence of a donut-shaped radiation pattern dominated by an in-plane momentum of $0.1 k_0$ (where $k_0$ represents the maximum collecting momentum) This observation indicates nearly collimated output in the normal direction. The central singularity in the donut-shaped pattern arises due to the non-radiative nature of the emission at $\Gamma$ point in the BIC.

Remarkably, the simulation result of Fig. 4c matches well with the measured BFP image (see discussion in Supplementary Note 5). Furthermore, the presence of polarization singularities in BIC polariton condensates is confirmed through a full polarization-resolved measurement of BFP images (Fig. 4d), along with the corresponding Stokes parameters $S_1$−$S_3$ maps (Fig. 4e, see Methods section). These measurements reveal the vortex profile and the unambiguous singularity point at $\Gamma$ point. The polarization orientations around $\Gamma$ point align with a vortex of topological charge $l = -1$[23]. Additionally, the phase singularity of the BIC polariton condensate emission is further demonstrated by the self-interference measurement based on the same Michelson interferometer with momentum-space images (Fig. 4f and

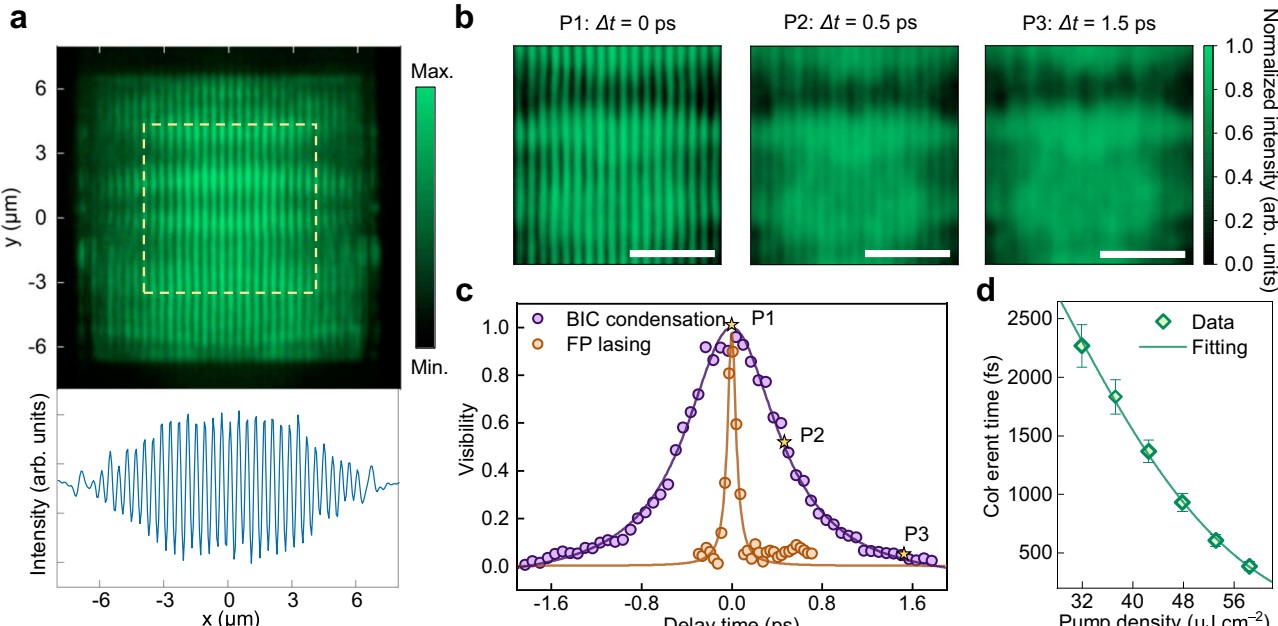

**Fig. 3 | Spatial and temporal coherence of BIC polariton condensate. a** Top, interference patterns of BIC polariton condensates in a typical CsPbBr$_3$ PhC lattice after the superposition of two mirror-symmetric images by the Michelson interferometer. Bottom, line profile of the coherence intensity extracted from the interference pattern (along $y = 0$). **b** Interference patterns (the dashed region in **a**) acquired at different time delays ($\Delta t$) of 0, 0.5, and 1.5 ps (denoted as P1, P2, and P3), respectively. The emission intensity at $\Delta t = 0$ was normalized for clarity. **c** Visibility of interference fringe as a function of delay time for BIC polariton condensation (purple) and the photonic lasing (orange). The visibility is obtained from time-dependent coherence fringes through the Fourier transform and fitted with the Gaussian function. Coherence times of 2.14 ps and 0.31 ps could be obtained for the BIC polariton condensation and the referenced pump laser, respectively. The yellow stars correspond to the three situations (P1, P2, and P3) in **b**. **d** Coherence time as a function of the pump density. The dots are taken from the measured temporal coherence for each value of pump density. The fitting line is calculated from power-dependent temporal coherence with an exponential function. The decrease in coherence time with increasing pump density is attributed to polariton–polariton scattering, which induces phase decoherence within the polariton lasing mode.

Supplementary Fig. S15). As the momentum-space images from the left arm and its inversion counterpart in the horizontal coordinates from the right arm become progressively misaligned with increasing distances, the two fork-shaped stripes remain while the number of fringes between them gets more (Supplementary Fig. S16), confirming the existence of orbital angular momentum in the vortex radiation with a topological charge of $l = -1$, inherited from the BIC mode[4,47,48].

Besides, the finite lateral dimension of the CsPbBr$_3$ PhC lattice and the interface between different optical modes and/or optical bandgap contribute to light trapping in the transverse direction, leading to the formation of discrete photonic modes. Accordingly, the BIC modes can be transformed into a series of localized modes M$_{pq}$ ($p$, $q$ are integers) in the transverse direction, which have recently been referred to as "miniaturized BIC"[49,50]. We experimentally verified the existence of these modes and successfully switched between them by applying a signal beam and a gate beam with different time delays (Fig. 5a), which is ubiquitous for polariton condensates due to the pronounced nonlinear interactions. The profile of the gating beam is set to be half the width of the PhC lattice to enhance the switching performance (Supplementary Figs. S17, 18). The switching of the miniaturized BIC polaritonic mode M$_{12}$ is achieved through polariton–polariton scattering at the fundamental M$_{11}$ mode, which is ultimately determined by the temporal evolution of the condensed BIC polariton density (Fig. 5b). The M$_{12}$ mode undergoes an on/off transition when the delay time is shifted from negative to positive (left panel of Fig. 5c). The far-field image of M$_{12}$ reveals highly directional emission at the four corners while retaining the central hollow characteristic of M$_{11}$ (right panel of Fig. 5c). Moreover, the measured Stokes parameters provide clear evidence that the polarization singularity at $\Gamma$ point is preserved in both the M$_{11}$ and M$_{12}$ modes (Fig. 5d). No additional singularity is observed in M$_{12}$, suggesting that M$_{12}$ remains a symmetry-protected BIC polariton mode rather than accidental BIC. Additionally, Fig. 5e

illustrates the temporal evolution of M$_{12}$ at various delay times. The intensity of M$_{12}$ undergoes a rapid reduction near zero-time delay, corresponding to a decrease in intensity by 12.5 dB within a time span of 17 ps. The corresponding far-field pattern also demonstrates the transition from M$_{12}$-dominated emission to the fundamental M$_{11}$ mode at positive delay times. This switching of miniaturized BIC polariton modes, exhibiting different energy levels while maintaining the identical topology, holds great promise for on-chip communication with enhanced light–matter interaction.

In conclusion, we demonstrate room-temperature BIC polariton condensation in perovskite air-hole PhCs. The utilization of BIC states provides a high-quality cavity, suppressing radiative losses and facilitating polariton accumulation. The resulting BIC polariton condensates exhibit long-range coherence and emit coherent vortex beams with low divergence angles in the vertical direction. Additionally, we realize switching between multiple orders of BIC polariton modes, while preserving their topological properties, in a miniaturized structure through the nonlinear interactions of polariton condensates. Our findings provide a pathway to achieve room-temperature coherent polariton condensates with orbital angular momentum, unlocking new possibilities for the utilization of integrated polaritonic devices at room temperature with enriched degrees of freedom.

## Methods

### Materials

Single-crystalline CsPbBr$_3$ microplatelets were fabricated by the chemical vapor deposition method. Initially, CsBr (99.99%, Sigma Aldrich) and PbBr$_2$ (99.99%, Sigma Aldrich) powder precursors were thoroughly mixed with a molar ratio of 1:1. The resulting mixture was then placed at the center of a quartz tube equipped with a heating furnace. Next, SiO$_2$/Si substrates were meticulously cleaned in an ultrasonic bath filled successively with ethanol, acetone, and deionized water for

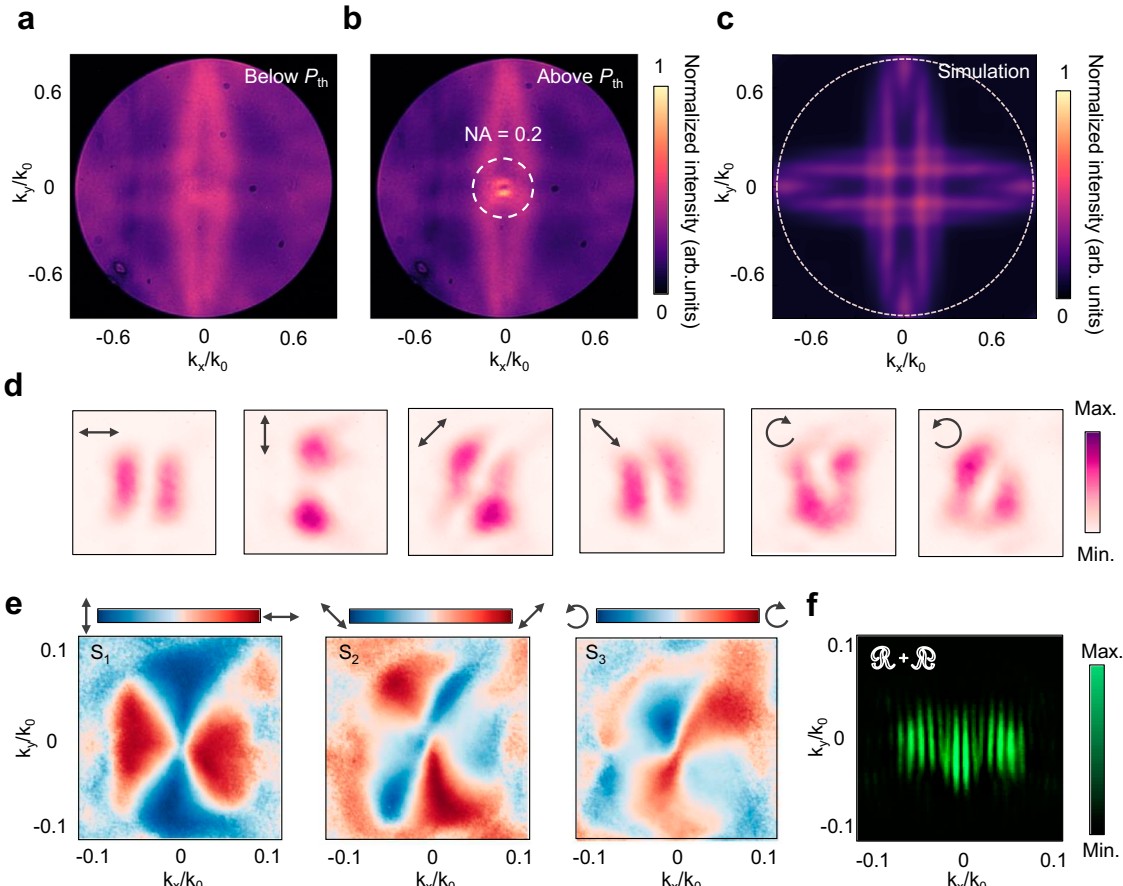

**Fig. 4 | Far-field emission characteristic of BIC polariton condensate.**
**a**, **b** Fourier space images of the PL emission excited with the pump density below
(**a**) and above $P_{th}$ (**b**). The dashed circle in **b** corresponds to an in-plane momentum
of $0.2k_0$, in which a donut-shaped BIC polariton condensate emission pattern is
presented. **c** The simulated Fourier space image corresponding to **a**. **d** Far-field
radiation patterns of the BIC polariton condensate vortex emission in six
representative polarization directions. **e** Stokes parameters $S_1$–$S_3$ map of the BIC
polariton condensate vortex emission above $P_{th}$. **f** Self-interference pattern of
Fourier space images observed by the Michelson interferometer. Pairs of mirror-
symmetric fork-shaped stripes reveal the nature of orbital angular momentum in
BIC polariton condensate vortex emission.

15 min each, then dried with nitrogen gas, and subsequently placed
downstream of the precursors. To ensure a controlled growth envir-
onment, the chamber was purged with high-purity nitrogen and
vacuumed to a pressure of 0.5 Pa. The mixed precursors were heated
to 575 °C and maintained at this temperature for 10 minutes, carried by
high-purity nitrogen flow under a pressure of 200 Torr and flux rate of
40 sccm. Finally, the entire facility cooled naturally to room
temperature.

### Fabrications
The periodic air-hole CsPbBr$_3$ PhC lattices were fabricated by FIB
milling. Using the FEI Nova 200 NanoLab FIB system, $10 \times 10\,\mu m^2$ arrays
of periodic air holes were patterned on the CsPbBr$_3$ microplatelet. The
nominal ion-beam current was controlled below 10 pA for a suitable
spot size of etching. Note that because perovskite is sensitive to the
damage of ion etching, etching paths of ion-beam scanning on the
perovskite microplatelets were optimized using the stream file func-
tion in the FEI system, and the fabrication dwell time was set as a
maximum of 2 ms to minimize damages in repetition.

### Morphology and structure characterizations
SEM images were conducted using Zeiss GEMINI II. X-ray measure-
ments were performed on a Rigaku SmartLab diffractometer using
CuK$_\alpha$ radiation ($\lambda = 1.5406$ Å). Atomic force microscope images were
obtained using a BRUKER Dimension Icon.

### Optical characterizations
The absorption spectrum of individual CsPbBr$_3$ microplatelet was
obtained by a home-built micro transmission/absorption spectro-
meter, which has been described in our previous work[51]. To obtain
angle-resolved reflection and PL spectra, a custom-made Fourier
imaging system was utilized, incorporating an objective lens (×50,
numerical aperture of 0.8) to focus light and improve spatial resolu-
tion. In reflection measurement, the white light beam from a tungsten-
halogen light source (SLS201L, Thorlabs) was focused with the objec-
tive lens onto the sample. The reflected light was collected to a Horiba
iHR-550 spectrometer with a liquid nitrogen-cooled charge-coupled
device detector and a grating of 1800 grooves per mm. For pulsed
laser excitation, an 800 nm pulse laser beam generated from a
Coherent Vitara-s oscillator (35 fs, 80 MHz), which was seeded by the
Coherent Astrella amplifier (80 fs, 1 kHz), was doubled to 400 nm with
a pulse duration of 300 fs and repetition rates of 1 kHz through a
barium boron oxide crystal. The pump spot was modulated by placing
an adjustable rectangular aperture followed by a lens of 300 mm focal
length before entering the Fourier system. In addition, two pump
sources with adjustable delay time were realized by a beam splitter and
retroreflectors coupled with a delay stage. A schematic of the optical
system setup is provided in Supplementary Fig. S4.

Time-resolved PL spectrum was conducted by a time-correlated
single photon counting (TCSPC, SPC-150) system (below condensed
threshold) and a home-built ultrafast optical Kerr gating (OKG) system

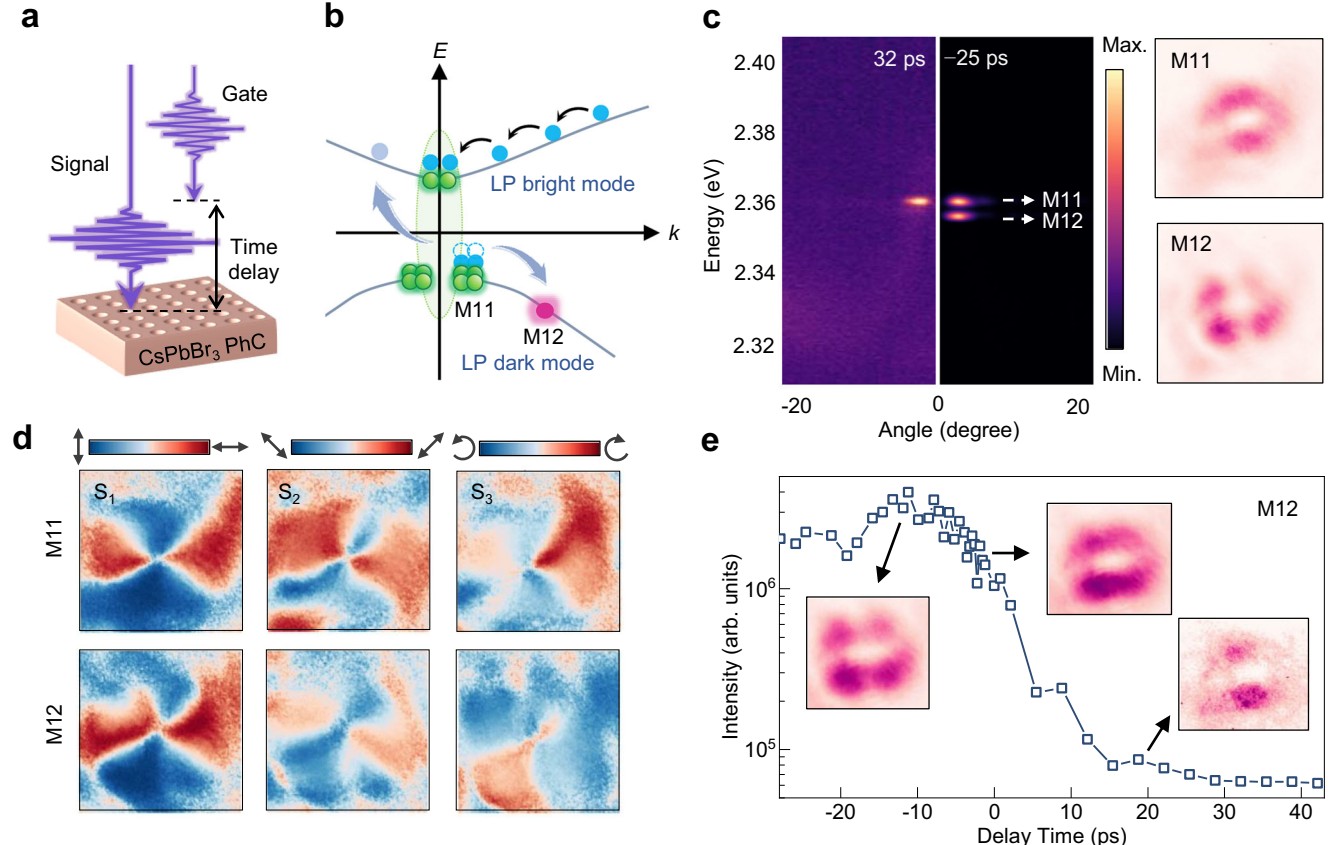

**Fig. 5 | Efficient switching for the miniaturized BIC polariton modes.**
**a** Schematic of the experimental setup. A signal beam ($-P_{th}$) and a gate beam ($-0.2$ $P_{th}$) with different time delays are applied. The gating beam excited half the width of the PhC lattice, while the signal beam excited the whole CsPbBr$_3$ PhC lattice. **b** Schematic of the switching mechanism of the miniaturized BIC polariton modes. **c** Left panel: angle-resolved PL spectra of the miniaturized BIC polariton (above condensed threshold). Right panel: the far-field momentum-space images of $M_{11}$ and $M_{12}$ modes. **d** Stokes parameters $S_1$–$S_3$ map of $M_{11}$ and $M_{12}$ modes. **e** Intensity as a function of the time delay for the $M_{12}$ mode. Inset: the corresponding far-field momentum-space images at time delays of $-25$, 0, and 15 ps, respectively.

(above condensed threshold). In detail, the TCSPC system was carried by an excitation source of a 400 nm, 80 MHz laser from Ti: sapphire oscillator (Chameleon Ultra, Coherent) and focused onto the sample by a ×100 objective (numerical aperture of 0.9) with a temporal resolution of ~40 ps. The time and angle-resolved PL is developed based on an optical Kerr gate coupled with the Fourier imaging system. The femtosecond pulses of 35 fs @ 800 nm, as mentioned previously, are divided into two parts. One part is frequency doubled to generate femtosecond pulses at the central wavelength of 400 nm. The sample is excited non-resonantly by focusing excitation pulses at 400 nm onto it, which leads to the excitation of polariton condensates. The remaining femtosecond pulses at 800 nm are precisely delayed relative to the UV excitation pulses and focused onto the Kerr medium. By varying the delay time of the OKG, the angle-resolved distribution of polariton condensate emission can be visualized frame by frame with sub-picosecond resolution.

Coherence and self-interference measurements were carried out by a Michelson interferometer attached to the Fourier imaging system. The emission pattern was split into two arms, and the image in one of the arms was inverted upside-down with a retro-reflector while the image in the other arm was kept constant. Then the two images with mirror symmetry were superposed again with a beam splitter and overlapped at the image plane simultaneously. The interference patterns were collected by a CMOS camera (Sunny Optics) for further analysis.

Polarization measurement was carried out by the combination of a half-waveplate (linear polarization) or a quarter-waveplate (circular

polarization) with a linear polarizer before the entrance of the imaging camera. In this way, maps of four linear polarizations (horizontal $I_H$, vertical $I_V$, diagonal $I_D$, antidiagonal $I_A$) and two circular polarizations (right $I_R$ and left $I_L$) were obtained. Three Stokes parameters can be calculated as: $S_1 = \frac{I_H - I_V}{I_H + I_V}$, $S_2 = \frac{I_D - I_A}{I_D + I_A}$, and $S_3 = \frac{I_R - I_L}{I_R + I_L}$.

## Numerical simulation methods
We used the commercial software Lumerical FDTD to perform the mode dispersions and Fourier space distribution. All the geometric parameters in the simulations were taken from the SEM images. The wavelength-dependent complex permittivity of CsPbBr$_3$ was obtained from the literature[39]. Numerical simulation of angle-resolved reflection and far-field distribution are described in detail in Supplementary Notes 2 and 5.

## Data availability
All data to evaluate the conclusions are present in the manuscript, and the Supplementary Material. Raw data are available from the corresponding authors on request.

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

## Acknowledgements

Q.Z. acknowledges funding support from the Natural Science Foundation of Beijing Municipality (JQ21004), and the National Natural Science Foundation of China (52072006 and U23A2076). X.L. acknowledges funding support from the National Key Research and Development Program (2023YFA1507002), the National Science Foundation for

Distinguished Young Scholars of China (22325301), the National Natural Science Foundation of China (22073022, 11874130, 12074086, 12204123, and 22173025), China Postdoctoral Science Foundation (2022M710925), the Strategic Priority Research Program of Chinese Academy of Sciences (XDB36000000), Beijing Municipal Natural Science Foundation (1222030), and the CAS Instrument Development Project (Y950291). Q.X. acknowledges funding support from the National Natural Science Foundation of China (12020101003) and strong support from the State Key Laboratory of Low-Dimensional Quantum Physics at Tsinghua University.

## Author contributions

X.L. and Q.Z. conceived the idea and led the project. X.L., Q.Z., X.W. S.Z., and J.S. designed experiments. X.W., W.D. and Y.Z. produced the samples. X.W., S.Z., J.S., X.Z., Z.Zhu., C.J., Y.C. and Y.W. performed the optical spectroscopy. X.W., X.D., X.Z. and S.Z. conducted the numerical simulations. X.L., Q.Z., Y.Z., B.W., Q.X., X.W., S.Z., J.S., Y.Z., Z.Zhang, and Y.L. analyzed the data. X.L., Q.Z., Q.X., X.W., S.Z., J.S., X.Z. and X.D. prepared the manuscript. X.L., Q.Z., Q.X. supervised the studies. All the authors discussed the results and revised the manuscript.

## Competing interests

The authors declare no competing interests.
