## [Peer Review File · Nature Communications]

REVIEWER COMMENTS

Reviewer #1 (Remarks to the Author):

Review of

As noted in my previous review of this paper, the authors report the condensation of BIC polaritons in a perovskite material. I believe the evidence for BIC polariton formation is robust, and further, the evidence for the appearance of condensation is sufficiently convincing.

I agree with the comments of Referee #2 that this paper (for publication in Nature Communications) needs to specifically emphasize the features of BIC polariton condensation at room temperature, as well as some resulting new physics, given that room temperature BIC polaritons have already been previously observed (ref [23]). I think the improved data on condensation in this revised manuscript does this. For this reason, I think this paper can eventually be published.

I think there are still some significant shortcomings to the paper however. Firstly, this does not seem to be a very high quality demonstration of a BIC polariton condensate, compared to e.g. GaAs BIC polaritons. BIC polariton condensation is surely more difficult in this material, and seems to be limited by sample size and the fabrication procedure, as highlighted by the authors.

Related to this, my key concerns, which I think can be addressed are (some still remaining from my original report):

- Relatively short lifetimes – BIC polaritons should have exceedingly long lifetimes. This is one of their most important features, and so I previously asked the authors to measure this, which they have done. The authors report lifetimes of 2.57ps and 3.07ps, which seems very short for such a structure. Meanwhile, GaAs is measured at 300ps (ref [6]). What is the lifetime of the bare exciton in this perovskite material? If the exciton lifetime in this material is much shorter than in GaAs, then this would partially explain the orders of magnitude difference. However, the data in Figure S2 suggests that it is in the ns time scale. Could the authors comment on / analyze this?

- Interference measurement in Figure 3: I previously commented that the interference fringes were not visible in the earlier version. The new version addresses this by changing the scale of the plot, and the colorbar reads 'min' and 'max'. The interference fringes do appear at short time scales, but the degree of visibility is not obvious with this colorbar scaling. This should be replaced with absolute values, so that the correspondence with panel d is clear.

I also have some remaining concerns on the vortex data.

· The previous data did not show the vortex image in the second interferometric arm. The authors describe a new interferometer setup and the image is now shown in reciprocal space. I am not sure why this is done. They cite a paper [Nat. Phys. 7, 129, (2011)] which they say uses a similar interferometer geometry, but this is not true. This interferometer also images two vortex patterns (one for each arm) in coordinate space.

Further, the two indicated forks (4f) are now almost overlapping, and the lines sketched over the interference fringes obscure the details. Based on the emission pattern (d) it seems likely that this is a vortex state, but why are the two vortex images not separated by at least one fringe so they are clear? Noise in an interferometer can cause similar distortions, and it is difficult to conclude that this is not the case here.

Reviewer #2 (Remarks to the Author):

The quality of the revised manuscript has been substantially improved. The reviewer would like to have the authors address the following suggestions before a final recommendation is made.

1.The numerical simulation are done only with infinite BIC structures. The author should include some calculations with real finite periodicity structures.

2.The line width narrowing behaviors are quite different for the two cases in Fig.2e. The author should discuss and explain it further.

3.Fig.3c captions has a typo, the curve is not red color but orange color.

4.In Fig 3d's caption, the authors wrote "Polariton-polariton interactions lead to the dephasing of temporal coherence, showing a decreasing trend in the coherence times" This is a bit strange argument, why?

5. The contrast in Fig 4a and 4b are different, in comparison with Fig. 4c. The author should further process them for better comparison. This suggestion also apply to other SI figures related to far field emission patterns.

6. Optical birefringence typically exists in CsPbBr₃ should be discussed in the manuscript further. It seems the author did not observed two modes? Why?

Response Letter (NCOMMS-23-45853-T)

Reviewer 1

Overall comment:

As noted in my previous review of this paper, the authors report the condensation of BIC polaritons in a perovskite material. I believe the evidence for BIC polariton formation is robust, and further, the evidence for the appearance of condensation is sufficiently convincing. I agree with the comments of Referee #2 that this paper (for publication in Nature Communications) needs to specifically emphasize the features of BIC polariton condensation at room temperature, as well as some resulting new physics, given that room temperature BIC polaritons have already been previously observed (ref [23]). I think the improved data on condensation in this revised manuscript does this. For this reason, I think this paper can eventually be published.

Response:

We sincerely appreciate the reviewer's positive evaluation and also the excellent suggestions on our manuscript. We have addressed all the comments raised by the reviewer point-by-point in the following sections.

Comment 1:

I think there are still some significant shortcomings to the paper however. Firstly, this does not seem to be a very high quality demonstration of a BIC polariton condensate, compared to e.g. GaAs BIC polaritons. BIC polariton condensation is surely more difficult in this material, and seems to be limited by sample size and the fabrication procedure, as highlighted by the authors. Related to this, my key concerns, which I think can be addressed are (some still remaining from my original report): Relatively short lifetimes – BIC polaritons should have exceedingly long lifetimes. This is one of their most important features, and so I previously asked the authors to measure this, which they have done. The authors report lifetimes of 2.57 ps and 3.07 ps, which seems very short for such a structure. Meanwhile, GaAs is measured at 300 ps (ref [6]). What is the lifetime of the bare exciton in this perovskite material? If the exciton lifetime in this material is much shorter than in GaAs, then this would partially explain the orders of magnitude difference. However, the data in Figure S2 suggests that it is in the ns time scale. Could the authors comment on / analyze this?

Response 1:

We thank the reviewer for pointing out this important issue.

In ref [6] [*Nature* 2022, **605**, 447-452], the cavity loss was identified as zero because the direct radiation of excitons into free space is significantly suppressed in BIC structures, thus, the BIC polariton lifetime is limited by the excitonic non-radiative lifetime, and the excitonic fraction. The

losses of polaritonic quasi-BIC were given as $\gamma_{\text{quasi-BIC}} = W_{+,LP} \times \gamma_{\text{nr}}$, where γ_{nr} is the nonradiative decay rate of an exciton, and $W_{+,LP}$ is the excitonic fraction. By considering a 50% excitonic fraction, and an excitonic non-radiative lifetime of approximately 150 ps for GaAs quantum wells at 4 K, a BIC polariton lifetime was obtained of ~ 300 ps.

However, in our CsPbBr₃ PhC lattices, we should not ignore the cavity photon out-coupling part considering the limited cavity quality factor in practice, which is inevitable by etching perovskite materials. The polariton lifetime should be expressed as the inverse to the population decay rate, that is $\gamma_{LP} = W_{+,LP} \cdot \gamma_{\text{nr}} + (1 - W_{+,LP}) \cdot \gamma_C$, where γ_C is the out-coupling rate of a cavity photon [ref¹: *Rev. Mod. Phys.* 2010, **82**, 1489-1537]. According to the estimation, a quality factor $<10^4$ is reasonable near BIC state energy, resulting in a calculated cavity photon lifetime of approximately 2.5 ps. It should be noted that the quality factor obtained from the angle-resolved reflectance spectra may underestimate the exact cavity quality factor. On the other hand, for CsPbBr₃ perovskites excited by femtosecond pulsed laser at room temperature, the excitonic non-radiative recombination processes may involve polaron-assisted energy transfer (ref²: <20 ps, *ACS Appl. Mater. Interfaces* 2021, **13**, 6820-6829), Auger recombination (ref³: <20 ps, *Angew. Chem. Int. Ed.* 2020, **59**, 14292-14295), and exciton-exciton annihilation (ref⁴: <60 ps, *Adv. Opt. Mater.* 2016, **4**, 1993-1997), etc. Therefore, considering a $W_{+,LP}$ of ~ 0.20 , and an excitonic non-radiative lifetime of ~ 60 ps, a polariton lifetime of ~ 3.1 ps could be estimated, which agrees with our experimental results.

In conclusion, the BIC polariton lifetime is not constrained by the excitonic radiative lifetime; instead, it is predominantly limited by the finite cavity quality in our CsPbBr₃ PhC lattices. Enhancing the quality factor of the cavity emerges as the pivotal factor for extending the room-temperature BIC polariton lifetime.

Revision 1:

Main text, Page 5: “The BIC polariton lifetime is predominantly limited by the finite cavity quality factor, which could be further alleviated through optimized fabrication processes (see discussion in Supplementary Note 4).”

In the **Supplementary information** part, added the following discussions: “Furthermore, the BIC polariton lifetime can be expressed as the inverse to the population decay rate, that is¹⁴, $\gamma_{LP} = |X|^2 \cdot \gamma_{\text{nr}}$

+ (1 - $|X|^2$) $\cdot\gamma_c$. Here, γ_c is the out-coupling rate of a cavity photon, γ_{nr} is the nonradiative decay rate of an exciton, and $|X|^2$ is the excitonic fraction. A practical quality factor $<10^4$ is reasonable near BIC state energy, resulting in a cavity photon lifetime of approximately 2.5 ps. It should be noted that the quality factor obtained from the angle-resolved reflectance spectra may underestimate the exact cavity quality factor. On the other hand, for CsPbBr₃ perovskites excited by femtosecond pulsed laser at room temperature, the excitonic non-radiative recombination processes may involve polaron-assisted energy transfer, Auger recombination, and exciton-exciton annihilation, *etc.* Hence, in our case, we should not ignore the cavity photon out-coupling part considering the limited cavity quality factor in practice. Then, considering a $|X|^2$ of ~ 0.20 and an excitonic non-radiative lifetime of ~ 60 ps, a polariton lifetime of ~ 3.1 ps could be estimated, which agrees with our experimental results.”

Comment 2:

Interference measurement in Figure 3: I previously commented that the interference fringes were not visible in the earlier version. The new version addresses this by changing the scale of the plot, and the color bar reads ‘min’ and ‘max’. The interference fringes do appear at short time scales, but the degree of visibility is not obvious with this color bar scaling. This should be replaced with absolute values, so that the correspondence with panel d is clear.

Response 2:

We thank the reviewer very much for the constructive suggestion. Following the reviewer’s comment, we now add the color bar scaling with normalized absolute values (Fig. R1). Furthermore, through optimization of material quality and fabrication processes, we fabricated samples demonstrating BIC polariton condensation with longer coherent times, evidenced by clear interference fringes at $\Delta t = 1.8$ ps (Fig. R2).

Fig. R1. a-c, Michelson interference patterns acquired at different time delays (Δt) of 0, 0.5, and 1.5 ps, respectively. The emission intensity at $\Delta t = 0$ was normalized for clarity.

Fig. R2. Michelson interference patterns of another sample acquired at $\Delta t = 0, 0.6, 1.2,$ and 1.8 ps, respectively.

Revision 2:

Main text, Page 5: “As the delay time extended from 0 to 1.5 ps (P1–P3), the fringes gradually blurred (Fig. 3b).”, and “...Note that longer coherence time can be achieved by further optimization of the material quality and fabrication processes (Supplementary Fig. S13).”

Revised Fig. 3 in the main text (shown below), interference patterns with three delay times (P1, P2, and P3) presented in Fig. 3b are highlighted with yellow stars in Fig. 3c for clarity:

Fig. 3 caption: “b, Interference patterns (the dashed region in a) acquired at different time delays (Δt) of 0, 0.5, and 1.5 ps (denoted as P1, P2, and P3), respectively.”, and “...The yellow stars correspond to the three situations (P1, P2, and P3) in (b).”

Supplementary Information: Add Fig. R2 as a new Supplementary Fig. S13.

Comment 3:

I also have some remaining concerns on the vortex data. The previous data did not show the vortex image in the second interferometric arm. The authors describe a new interferometer setup and the image is now shown in reciprocal space. I am not sure why this is done. They cite a paper [Nat.

Phys. 7, 129, (2011)] which they say uses a similar interferometer geometry, but this is not true. This interferometer also images two vortex patterns (one for each arm) in coordinate space.

Response 3:

We thank the reviewer very much for pointing out this issue. The Michelson interferometer setup employed is illustrated in Fig. R3a. The difference between our setup and the configuration mentioned in *Nat. Phys.* 2011, 7, 129 is that we superposition two momentum-space images instead of real-space images for interference pattern. The k -space image (left arm, Fig. R3b) overlaps with its inversion counterpart in the horizontal coordinates (right arm, Fig. R3c), resulting in the interference pattern (Figs. R3d-e).

Fig. R3. a, The experimental setup of the Michelson interferometer integrated into the Fourier system. The superposition of the original image and its inverted counterpart results in a mirror-symmetrical interference image. **b-c**, Back-focal plane (BFP) images of BIC polariton condensate emission collected from the left (**b**) and right arm (**c**) of the interferometer. **d**, Michelson interference pattern obtained by overlapping (**b**) and (**c**). **e**, Magnified interference pattern from (**d**), where two forks exhibiting a mirror-symmetric configuration are observed.

Revision 3:

Main text, Page 6: “...Additionally, the phase singularity of the BIC polariton condensate emission is further demonstrated by the self-interference measurement based on the same Michelson interferometer with momentum-space images (Fig. 4f and Supplementary Fig. S14)...”

Supplementary Information: Add Fig. R3 as a new Supplementary Fig. S14.

Comment 4:

Further, the two indicated forks (4f) are now almost overlapping, and the lines sketched over the interference fringes obscure the details. Based on the emission pattern (d) it seems likely that this is a vortex state, but why are the two vortex images not separated by at least one fringe so they are clear? Noise in an interferometer can cause similar distortions, and it is difficult to conclude that this is not the case here.

Response 4:

Thank you very much for pointing out this issue. As the k -space images from the left arm and its inversion counterpart in the horizontal coordinates from the right arm become progressively misaligned, with increasing distances ($0 < \Delta x_1 < \Delta x_2 < \Delta x_3$), the number of fringes between the two forks gets more [ref.⁵ *Phys. Scr.* 2019, **94**, 055502] (Fig. R4). Thus, the potential of noise in the interferometer can be ruled out.

Fig. R4. BFP images of BIC polariton condensate emission collected from the left arm (first column), the right arm of the interferometer (second column), and their interference pattern (third column), with distinct misaligned distances ($0 < \Delta x_1 < \Delta x_2 < \Delta x_3$).

Revision 4:

Main text, Page 7: “...As the momentum-space images from the left arm and its inversion counterpart in the horizontal coordinates from the right arm become progressively misaligned with increasing distances, the two fork-shaped stripes remain while the number of fringes between them gets more (Supplementary Fig. S15), confirming the existence of orbital angular momentum in the vortex radiation with a topological charge of $l = -1$, inherited from the BIC mode^{4, 47, 48}.”

Revised Fig. 4 in the main text:

Supplementary Information: Add Fig. R4 as a new Supplementary Fig. S15.

Reviewer 2

Overall comment:

The quality of the revised manuscript has been substantially improved. The reviewer would like to have the authors address the following suggestions before a final recommendation is made.

Response:

We sincerely appreciate the reviewer's positive evaluation, and we have addressed the six questions raised below.

Comment 1:

The numerical simulation are done only with infinite BIC structures. The author should include some calculations with real finite periodicity structures.

Response 1:

We express our gratitude to the reviewer for identifying the issue related to the lack of calculations of the actual finite periodicity. In response to the impact of the finite size of structure on BIC modes, we conducted computational investigations by the FDTD method, concerning mainly the varying number of periods (N). We introduced several finite values of N ranging from 5 to 40. We first made a simulation of a referenced sample for comparison with different finite-size parameters by FDTD. The calculated parameters are set as thickness $h = 140$ nm, period $a = 292$ nm, and radius of etched hole $r = 57$ nm. The angle-resolved reflectance spectra of the perovskite PhC lattice are depicted in Fig. R5. As the parameter N gradually increases from 5 to infinity, the TM modes of the CsPbBr₃ PhC lattice begin to emerge and exhibit a progressively narrowing mode linewidth. Particularly noteworthy is the observation that when N attains a value of 40, closely resembling practical dimensions in experimental scenarios, the resulting behavior closely mimics that of an infinite structure, as illustrated in Fig. R5e.

Fig. R5. Simulated angle-resolved reflectance spectra of CsPbBr₃ PhC lattices with different number of periods (N). a-d, Simulated angle-resolved reflectance spectra of CsPbBr₃ PhC lattice with N values of 5, 10, 20, and 40, respectively. e, Simulated angle-resolved reflectance spectra of CsPbBr₃ PhC lattice with an infinite number of periods.

Revision 1:

Main text, Page 4: “...In addition, the air-hole PhC lattice, fabricated with 40 periods, exhibits a narrow mode linewidth closely resembling that of an infinite structure (Supplementary Fig. S5)...”

Supplementary Information: Add Fig. R5 as a new Supplementary Fig. S5.

Comment 2:

The line width narrowing behaviors are quite different for the two cases in Fig.2e. The author should discuss and explain it further.

Response 2:

We thank the reviewer for pointing out this issue. We re-checked the linewidth extraction process for the two samples presented in Fig. 2e with different detuning, and found that they actually possessed similar linewidth narrowing behaviors (Fig. R6).

Fig. R6. Linewidth as a function of the pump density for two samples with detuning energies of -74.6 and -64.1 meV, respectively. The error bar represents the standard deviation.

Our analysis revealed potential discrepancies in the previous data extraction process, thus we now fix them and add an error bar. Specifically, numerous data points were measured at the threshold power closely preceding the appearance of BIC mode. Initially, excited BIC mode exhibits a closer proximity to the adjacent fluorescence peak, resulting in reduced contrast. As a consequence, the linewidth of these data points displays relatively higher relative errors during the

actual extraction process. Moreover, the proportion of these data points concerning the overall effective data points is comparatively significant, leading to an excessive amplification of minor variations in the linewidth narrowing process.

Moreover, we have explored the linewidth versus pump density in more samples with different detuning (Fig. R7). While slight variations were observed due to inherent differences between individual samples, we can confidently assert that the observed linewidth narrowing behavior remains robust across the entire dataset.

Fig. R7. Linewidth as a function of the pump density for the other two samples with detuning energies of -89.6 and -114.4 meV, respectively. Note that the perovskite microplatelets were fabricated separately, and the FIB milling process was conducted in a distinct batch. Consequently, direct comparisons of the condensation threshold are challenging due to potential variations introduced during different fabrication and processing stages.

Revision 2:

Main text, Fig. 4: revised Fig. R6 as a new Fig. 2e.

Comment 3:

Fig.3c captions has a typo, the curve is not red color but orange color.

Response 3:

We thank the reviewer very much for pointing out this mistake. Following the reviewer's comment, we have changed the word "red" to "orange" in the main text.

Revision 3:

Main text, Fig. 3 caption: "Visibility of interference fringe as a function of delay time for BIC polariton condensation (purple) and the photonic lasing (orange)..."

Comment 4:

In Fig 3d's caption, the authors wrote "Polariton–polariton interactions lead to the dephasing of temporal coherence, showing a decreasing trend in the coherence times" This is a bit strange argument, why?

Response 4:

We appreciate the reviewer for pointing out the inaccuracy of the prior description regarding “the dephasing of temporal coherence”. Specifically, at higher pump levels ($3P_{th}$ – $6P_{th}$ presented in Fig. 3d), a substantial reduction in coherence time occurs due to polariton–polariton scattering, inducing phase decoherence within the polariton lasing mode, which is well demonstrated both experimentally and theoretically in GaAs quantum wells polaritonic systems [ref.^{6,7}: *Phys. Rev. B* **85**, 205310 (2012), and *Phys. Rev. X* **6**, 011026 (2016)]. Meanwhile, this trend aligns with experimental observations of the broadening of emission linewidth and an accelerated decay rate at higher pump densities, providing another evidence of the occurrence of BIC polariton condensation. Following the reviewer’s comment, we have revised the description for clarity in the main text.

Revision 4:

Main text, Page 6: “...Fig. 3d reveals a decreasing trend of temporal coherence with increasing pump density, attributed to phase decoherence within the polariton lasing mode resulting from enhanced polariton–polariton scattering^{43, 46}.”

Fig. 3d caption: “...The decrease in coherence time with increasing pump density is attributed to polariton–polariton scattering, which induces phase decoherence within the polariton lasing mode.”

Comment 5:

The contrast in Fig 4a and 4b are different, in comparison with Fig. 4c. The author should further process them for better comparison. This suggestion also apply to other SI figures related to far field emission patterns.

Response 5:

We thank the reviewer for the constructive suggestion. Following the reviewer’s comment, we have modified the color range of the simulation result in Fig. 4c to match the experimental result presented in Fig. 4a (Fig. R8). It should be noted that, since some of the excitons not coupled with the BIC modes exist, the experimental Fourier space shown in Fig. R8a possesses an unavoidable

background emission, but we can still distinguish that the polaritonic BIC modes which are in line with the simulation results in Fig. R8b. We also incorporated a color bar with normalized finite values for Supplementary Fig. 25 for better comparison (Fig. R9).

Fig. R8. **a**, Fourier space images of the PL emission excited with the pump density below P_{th} . **b**, The simulated Fourier space image corresponding to (a).

Fig. R9. Simulated far-field distribution of electronic field intensity at different wavelengths.

Revision 5:

Main text, Fig. 4: revised Fig. R8b as a new Fig. 4c.

Supplementary Information: revised Fig. R9 as a new Supplementary Fig. S29.

Comment 6:

Optical birefringence typically exists in CsPbBr₃ should be discussed in the manuscript further. It seems the author did not observe two modes? Why?

Response 6:

We thank the reviewer very much for pointing out this issue. As noted by the reviewer, the observation of BIC polariton condensation is distinctly evident in the TM-polarization, as depicted in Figs. R10a-c. To further elucidate this observation, we performed simulations considering the TM- and TE-polarization reflectance spectra, as illustrated in Figs. R10d-g. It is noteworthy that, without accounting for the optical birefringence of CsPbBr₃ in our simulations, the TM-polarization reflectance spectra exhibit a close match between experimental and simulated results. However, an energy difference is evident between the experimental and simulated TE-polarization reflectance spectra, which can be attributed to the optical birefringence inherent in CsPbBr₃. Moreover, as shown in Figs. R10h-i, the observation of BIC polariton condensation exclusively in the TM-polarization can be attributed to the longer lifetime of the TM-polarized state, stemming from a higher excitonic fraction (~16%), in contrast to the TE-polarized state (~1%) [ref⁸: *Nano Lett.* **21**, 3120-3126 (2021)].

Fig. R10. Mode splitting resulting from the optical birefringence of the orthorhombic CsPbBr₃ single crystals. a-b, Angle-resolved reflectance spectrum of the air-hole CsPbBr₃ PhC

lattice. A linear polarizer was incorporated into the collection path to extract between emissions of TM-polarization (TM-pol, **a**) and TE-polarization (TE-pol, **b**), respectively. **c-d**, The corresponding simulations without considering optical birefringence of CsPbBr₃. **e-f**, The corresponding simulated excitonic weights based on the coupled harmonic oscillator model. The colors in the images represent a linear representation of the excitonic fraction for each mode, ranging from 0 (photon) to 1 (exciton). The observation of BIC polariton condensation exclusively in the TM-polarization can be attributed to the longer lifetime of the TM-polarized state, stemming from a higher excitonic fraction (~16%), in contrast to the TE-polarized state (~1%).

Revision 6:

Main text, Page 5: "...Additionally, despite the ubiquitous optical birefringence in orthorhombic CsPbBr₃ single crystals, BIC polariton condensation can be only observed in TM polarization, which could be attributed to the longer lifetime of the TM-polarized state resulting from a higher excitonic fraction (Supplementary Fig. S10)."

Supplementary Information: Add Fig. R10d-i as a new Supplementary Fig. S10.

References:

1. Deng, H., Haug, H. & Yamamoto, Y. Exciton-polariton Bose-Einstein condensation. *Rev. Mod. Phys.* **82**, 1489-1537 (2010).
2. Zhu, Y. et al. Inhomogeneous trap-state-mediated ultrafast photocarrier dynamics in CsPbBr₃ microplates. *ACS Appl. Mater. Interfaces* **13**, 6820-6829 (2021).
3. Li, Y., Luo, X., Ding, T., Lu, X. & Wu, K. Size- and halide-dependent Auger recombination in lead halide perovskite nanocrystals. *Angew. Chem.-Int. Edit.* **59**, 14292-14295 (2020).
4. Wei, K., Zheng, X., Cheng, X., Shen, C. & Jiang, T. Observation of ultrafast exciton-exciton annihilation in CsPbBr₃ quantum dots. *Adv. Opt. Mater.* **4**, 1993-1997 (2016).
5. Lan, B. et al. The topological charge measurement of the vortex beam based on dislocation self-reference interferometry. *Phys. Scr.* **94**, 055502 (2019).
6. Haug, H., Doan, T.D., Cao, H.T. & Thoai, D.B.T. Temporal first- and second-order correlations in a polariton condensate. *Phys. Rev. B* **85**, 205310 (2012).
7. Kim, S. et al. Coherent polariton laser. *Phys. Rev. X* **6**, 011026 (2016).
8. Wu, J.Q. et al. Nonlinear parametric scattering of exciton polaritons in perovskite microcavities. *Nano Lett.* **21**, 3120-3126 (2021).

REVIEWERS' COMMENTS

Reviewer #1 (Remarks to the Author):

The authors have satisfactorily responded to my questions / comments. I recommend publication.

I also add some minor comments to improve the presentation quality of the paper:

Figure 2:

- panels d, h - the linear or log scaling should be made clear with some numbers on the axis, even though it is in a.u.
- The orange and blue lines in h are not presently described in the caption (or the text that I could find).

Figure S14:

The interferometer sketch in (a) is not very clear and has inaccuracies. The prism is misoriented. It is not also immediately clear what orientation the two forks from the two separate arms in the interferometer should have. This might be indicated by a letter such as 'R', and its appropriate reflection to show the orientation of the second arm.

Extraction of linewidths in Figure 2:

This is in reference to a comment to Referee #2. The authors note that there were some errors in the original extraction of the linewidth in the processed data. In light of this Response by the authors, to make the process of linewidth extraction transparent, I suggest to show some raw data in the Supplementary Information, e.g. as a waterfall plot or something similar, so that the narrowing, and its equivalence across separate measurements is clear (as described in the response to Referee #2). Linewidth extraction might be complicated by disorder or other processes which introduces multiple closely spaced peaks.

Reviewer #2 (Remarks to the Author):

The revised manuscript addressed all the previous concerns very well.

The last two minor changes to improve the quality of the manuscript are as follows:

1. During the review process, a similar publication with organic materials is in Nano Letters <https://pubs.acs.org/doi/10.1021/acs.nanolett.3c01102> titled "Room Temperature Exciton–Polariton Condensation in Silicon Metasurfaces Emerging from Bound States in the Continuum". The author should cite and discuss briefly this highly relevant work in the manuscript.
2. The author should remove the "Bose-Einstein" term in the title, due to the non-equilibrium nature of the polariton condensation

Response Letter (NCOMMS-23-45853A)

Reviewer 1

The authors have satisfactorily responded to my questions / comments. I recommend publication. I also add some minor comments to improve the presentation quality of the paper:

Comment 1:

Figure 2:

- panels d, h - the linear or log scaling should be made clear with some numbers on the axis, even though it is in a.u.

- The orange and blue lines in h are not presently described in the caption (or the text that I could find).

Response 1:

We thank the reviewer for pointing out these issues. Following the reviewer's comment, we now add numbers on the axis in Figure 2d and Figure 2h to clarify the log scaling, and we also include descriptions of the orange and blue lines in the figure caption.

Revision 1:

Main text, new Figures 2d and 2h:

Revised Figure 2d:

Revised Figure 2h:

Main text, Figure 2h caption: “**h**, Integrated emission intensity as a function of the pump density of CsPbBr₃ PhC lattices (left two blue lines) and pristine microplatelets (right three orange lines) with similar sizes under quasi-CW excitation at 80 K.”

Comment 2:

Figure S14:

The interferometer sketch in (a) is not very clear and has inaccuracies. The prism is misoriented. It is not also immediately clear what orientation the two forks from the two separate arms in the interferometer should have. This might be indicated by a letter such as 'R', and its appropriate reflection to show the orientation of the second arm.

Response 2:

We thank the reviewer for pointing out this issue. Following the reviewer's comment, we have revised the schematic of the Michelson interferometer in Figure S14a, and added necessary notations for clarity.

Revision 2:

Supplementary information, new Figure S14a:

Comment 3:

Extraction of linewidths in Figure 2: This is in reference to a comment to Referee #2. The authors note that there were some errors in the original extraction of the linewidth in the processed data. In light of this Response by the authors, to make the process of linewidth extraction transparent, I suggest to show some raw data in the Supplementary Information, e.g. as a waterfall plot or something similar, so that the narrowing, and its equivalence across separate measurements is clear (as described in the response to Referee #2). Linewidth extraction might be complicated by disorder or other processes which introduces multiple closely spaced peaks.

Response 3:

We thank the reviewer for the constructive suggestion. Following the reviewer's comment, we have added emission spectra with pump densities near the condensation threshold (P_{th}) to clearly present the linewidth narrowing feature.

Fig. R1. Evolution of emission spectra with different pump densities of CsPbBr₃ PhC lattices at different detunings $\Delta = -64.1$ (a) and -74.6 meV (b) in the vicinity of P_{th} .

Revision 3:

Supplementary Information: add Fig. R1 as a new Supplementary Fig. 8.

Reviewer 2

The revised manuscript addressed all the previous concerns very well. The last two minor changes to improve the quality of the manuscript are as follows:

Comment 1:

During the review process, a similar publication with organic materials is in Nano Letters <https://pubs.acs.org/doi/10.1021/acs.nanolett.3c01102> titled “Room Temperature Exciton–Polariton Condensation in Silicon Metasurfaces Emerging from Bound States in the Continuum”. The author should cite and discuss briefly this highly relevant work in the manuscript.

Response 1:

We thank the reviewer for pointing out this issue. Following the reviewer’s comment, we now cite this work and give a brief discussion.

Revision 1:

Main text, page 8: “It should be noted that recently, polariton condensation has also been demonstrated in organic materials combining with BICs in silicon metasurfaces⁴⁷, further underscoring the broad applicability of BIC polariton condensates at room temperature.”

Comment 2:

The author should remove the “Bose-Einstein” term in the title, due to the non-equilibrium nature of the polariton condensation.

Response 2:

We thank the reviewer for pointing out this issue. Following the reviewer’s comment, we have revised the title.

Revision 2:

New title: “Exciton Polariton Condensation from Bound States in the Continuum at Room Temperature”